# Molecular Methods for Identification and Quantification of Foodborne Pathogens

**DOI:** 10.3390/molecules27238262

**Published:** 2022-11-26

**Authors:** Min Zhang, Jiajia Wu, Zhaoai Shi, Aocheng Cao, Wensheng Fang, Dongdong Yan, Qiuxia Wang, Yuan Li

**Affiliations:** 1Institute of Plant Protection, Chinese Academy of Agricultural Sciences, Beijing 100193, China; 2Beijing Innovation Consortium of Agriculture Research System, Beijing 100193, China

**Keywords:** polymerase chain reaction, isothermal amplification, loop-mediated isothermal amplification, nucleic acid sequence-based amplification, gene chip, gene probe technology

## Abstract

Foodborne pathogens that enter the human food chain are a significant threat worldwide to human health. Timely and cost-effective detection of them became challenging for many countries that want to improve their detection and control of foodborne illness. We summarize simple, rapid, specific, and highly effective molecular technology that is used to detect and identify foodborne pathogens, including polymerase chain reaction, isothermal amplification, loop-mediated isothermal amplification, nucleic acid sequence-based amplification, as well as gene chip and gene probe technology. The principles of their operation, the research supporting their application, and the advantages and disadvantages of each technology are summarized.

## 1. Introduction

Food free from harmful contaminants is vital for the well-being of humans. Food safety and human health are therefore priorities in most countries. Foodborne pathogens can contaminate food supplies leading to sudden and unexpected food outbreaks. Preventive measures, such as health publicity and education, effective import control, continuous food inspection, and good sanitation in manufacturing facilities often need to be supplemented with effective food monitoring and surveillance procedures. Those procedures must be capable of detecting pathogens before or shortly after they enter the food chain in order to isolate the source of the infection before it spreads to the wider population.

According to WHO, foodborne pathogens that contaminate human food and water became a major cause of human illness and mortality globally [1]. Accidental consumption of them, either directly from livestock and poultry or directly from plant-based food, can result in food poisoning. Common foodborne pathogens include *Salmonella* spp., *Staphylococcus aureus*, *Bacillus cereus*, *Vibrio parahaemolyticus, Campylobacter* spp., and *Shigella* spp. [2,3].

In the United States, approximately 76 million people became infected annually with foodborne diseases. Outbreaks of foodborne diseases increased recently in China [4], in the European Union (EU) [5,6,7], and in Bari. The detection and management of foodborne pathogens are a crucially important step for preventing the disease, illness, and mortality they cause.

Traditional microbiological quantification techniques rely on the culture of the pathogens on specialized media, their isolation, followed by biochemical identification, which generally takes about 5~7 days, during which time the pathogens are spreading further from the source of contamination making effective management of them difficult. The development and implementation of rapid, simple, sensitive, and species-specific methods for detecting foodborne pathogens is urgently needed in the food industry and by public health protection agencies that aim to maximize the health and safety of consumers. 

Recently developed technologies can be summarized into two categories, namely molecular methods and biosensors. Biosensors have benefits over conventional approaches by being faster, more cost effective, easy to carry out, and less labor-intensive. However, the results of biosensors are not very reliable, and there might be a need to develop a specific sensor for each food or specific analytical tools and sampling methods [8,9]. Therefore, various quantitative and molecular methods were developed that are suitable for the detection of foodborne pathogens [10,11]. This review summarizes the principles and characteristics of those methods, discusses their advantages and disadvantages, and assesses their suitability for the detection and identification of foodborne pathogens.

## 2. Polymerase Chain Reaction

Polymerase chain reaction (PCR) is a molecular method developed more than 30 years ago (Mullins et al. 1986) [12] to rapidly increase copies of all or part of a DNA sequence specific to a particular pathogen that allows further analysis of that genetic sequence. As PCR can only detect a particular DNA sequence at a time, and there are often many pathogens in a contaminated food sample, many PCR methods were developed, including multiplex PCR (mPCR), nested PCR, reverse transcription PCR, and real-time fluorescent quantitative PCR (RT-qPCR). Our paper summarizes mPCR and RT-qPCR, as they are the most widely used for the pathogen detection and identification of foodborne pathogens.

### 2.1. Multiplex PCR

The advantages of mPCR are being highly species-specific, highly sensitive, and the capability of detecting different species of pathogenic organisms simultaneously [13]. This technology enables the rapid detection of multiple microorganisms in a single reaction that can simultaneously amplify multiple sites. The principle of this technology is that multiple pairs of primers present in the reaction mixture amplify different target gene fragments in parallel. mPCR is mainly used for gene knockout, mutation analysis, and RNA detection. Thereby, this technology can improve the health and safety of food, as shown in Figure 1.

As examples, Molina et al. (2015) [14] used mPCR to identify Escherichia coli in a foodborne disease outbreak by targeting the lacZ and yaiO genes. Rosimin et al. (2016) [15] used mPCR to rapidly identify Listeria monocytogenes in vegetables. Yang et al. (2021) [16] used mPCR together with membrane chip technology to simultaneously detect nine species of foodborne pathogens in food. Li et al. (2019) [17] used mPCR technology to detect the tlh, tdh and trh genes of *Vibrio parahaemolyticus*. In this study, the prevalence rate of *Vibrio parahaemolyticus* was 19%, indicating that this bacterium was highly pathogenic.

mPCR can therefore identify many different species of pathogens that commonly contaminate food and that cause similar poisoning symptoms in humans. However, since the design of primers is the key factor in developing mPCR determination, there may be some interactions between multiple primer sets, resulting in low amplification efficiency. Therefore, primer sets should be designed with similar annealing temperature, and provide a method to distinguish amplicons after a thermal cycle [18]. Additionally, it can lead to a false positive result as living and dead bacteria cannot be distinguished [19]. Therefore, mPCR can often lead to unsatisfactory results.

### 2.2. Real-Time Fluorescent Quantitative PCR

RT-qPCR includes chemicals that fluoresce in the PCR reaction system. The presence of pathogenic DNA causes the mixture to fluoresce, thereby enabling pathogen presence to be monitored in real time. The level of fluoresce can be used to quantify the concentration of pathogenic organisms present in the sample when compared with a standard curve [20]. 

RT-qPCR is highly specific and sensitive. Amplified products are detected in real time without the need for post-PCR DNA analysis. For those reasons, RT-qPCR became one of the most preferred methods for detecting and identifying foodborne pathogens. The most commonly used RT-qPCR methods utilize TaqMan™ and LightCycler™ probes [21].

Tetsuya et al. (2015) [22] used RT-qPCR technology to detect three Stx1 subtypes and seven Sta2 subtypes linked to Shigella toxin. Nadin-Davis et al. (2018) [23] used RT-qPCR technology to detect Salmonella in samples obtained from 239 poultry facilities. 

Alia et al. (2020) [24] developed quadruple probe RT-qPCR technology to distinguish between four serotypes of Listeria monocytogenes found in processed and prepared meat products. The probes are highly sensitive, species-specific, and rapid. Their research results were important for minimizing the risk of L. monocytogenes contamination in meat products and for establishing the routine surveillance of persistent strains of that organism in ready-to-eat meat products. 

Zhang et al. (2020) [25] compared RT-qPCR technology with the national standard method used to simultaneously detect foodborne pathogens in 60 animal, aquatic, and dairy products. Although RT-qPCR technology detected pathogens at a higher rate than the national standard method, it failed to detect the important pathogen Staphylococcus aureus.

Martin et al. (2013) [26] detected Salmonella in cooked ham by RT-qPCR, and the results show that the detection limit was 103 CFU/g. Ma et al. (2013) [27] detected Staphylococcus aureus, Salmonella, and Shigella in the detection of fresh pork, and their detection time was no more than 8 h, which was shorter than the conventional PCR detection time. The research results show that the detection limit of Staphylococcus aureus was 9.6 CFU/g, that of Salmonella was 2.0 CFU/g, and that of Shigella was 6.8 CFU/g. Ranjba et al. (2016) [28] detected *E. coli* O157:H7, and the detection efficiency was greatly improved. The detection could be completed in less than 30 min. The results show that the detection limit was 78 pg/tube.

RT-qPCR and ordinary culture methods both produced consistent results, but RT-qPCR produced results in less time. As RT-qPCR technology is carried out in a closed system, false positives caused by contaminants at the time of analysis can be avoided. As electrophoresis is not required after amplification, the time to carry out RT-qPCR is much shorter than the standard method. However, the cost of RT-qPCR technology is high, the equipment is expensive, and multiple probe RT-qPCR is difficult to undertake, as operators require a high level of technical skill [29]. The summary of this part is shown in Table 1.

## 3. Isothermal Amplification

Isothermal amplification is a novel nucleic acid amplification method that provides a rapid, sensitive, accurate, and specific test for the detection of foodborne pathogens. Specialized and expensive equipment is not required, which is appealing to many food production industries. 

The main isothermal amplification technologies are loop-mediated isothermal amplification (LAMP) and nucleic acid sequence-based amplification (NASBA). 

### 3.1. Loop-Mediated Isothermal Amplification

Notomi et al. (2000) [32] first reported the molecular technique of nucleic acid amplification, termed loop-mediated isothermal amplification (LAMP), where a set of four (or six) different primers bind to six (or eight) different regions on the target gene, making it highly specific. All the reactions can be carried out under isothermal conditions ranging from 60 to 65 °C. Post-amplification electrophoresis is not required, as detection is simply by visual judgment with the unaided eye [33]. The process of LAMP is as shown in Figure 2.

The first application of LAMP for foodborne pathogens is to detect *stxA*_2_ in *Escherichia coli* O157:H7 cells. The mild permeability conditions and low isothermal temperature used in the in LAMP are less harmful than those caused by in situ PCR. The results show that the image contrast obtained by this method is higher than that of in situ PCR [34].

Xu et al. (2021) [35] used LAMP to detect common foodborne pathogens in dairy products. LAMP is very sensitive and rapid, as it can detect just a single copy of a gene within 30 min.

The LAMP method for rapid detection of the foodborne *Salmonella* strains was developed and evaluated by Zhao et al. (2010) [36] The optimal reaction condition was found to be 65 °C for 45 min, with the detection limit as 1 pg DNA/tube and 100 CFU/reaction. This application of LAMP assays was performed on 214 foodborne *Salmonella* strains using a rapid procedure and easy result confirmation, where the specificity of LAMP and PCR assays was 97.7% (209/214) and 91.6% (196/214), respectively; with a 100% specificity for both assays. Simultaneously, high specificity was acquired when the LAMP assay was subjected to 39 reference strains, with no false positive amplification observed. In conclusion, this LAMP assay was demonstrated to be a useful and powerful tool for rapid detection of *Salmonella* strains. Compared with previously reported LAMP assays for the detection of *Salmonella* strain, the improved LAMP method in the present study offers advantages on easiness in operation and time consumption. The total detection time, including DNA preparation, LAMP reaction, and results determination, was approximately 60 min. Undoubtedly, rapidness, easiness, and cost-effectiveness of LAMP assay will aid in the broad application of bacteriological detection of *Salmonella*.

LAMP is a relatively simple method for detecting specific pathogens. The amplification reaction can be completed at a constant temperature. LAMP is 1~2 orders of magnitude more sensitive than conventional PCR technology. The test can be undertaken in 30~60 min [37]. However, designing the primers is difficult and costly; the length of an amplified sequence cannot exceed 300 bp; non-specific pairing between loop primers will lead to false positives; and cross-contamination is likely unless precautions to avoid contamination are put in place [38].

### 3.2. Nucleic Acid Sequence-Based Amplification

Compared with traditional PCR that detects DNA, NASBA relies on the isothermal amplification of RNA in the target organism. Thermal cycling equipment is not required. NASBA contains a primer that binds to the target RNA sequence, and a cDNA strand is produced with reverse transcriptase. RNase H is then used to digest the template RNA and the cDNA is bound to a second primer for the production of double-stranded cDNA using reverse transcriptase. Finally, T7RNA polymerase is used to produce RNA transcripts via an amplification process. 

This method is particularly suited to the detection of RNA viruses because an RNA polymerase is used to amplify RNA without conversion to cDNA [39,40,41]. NASBA was proven to be a sensitive, specific, and rapid analysis method for the detection of several foodborne pathogens [42]. Please see the Table 2 below for specific examples.

NASBA is a sensitive transcription-based amplification system that uses a battery of three enzymes leading to a main amplification product of single-stranded RNA, and is specifically designed for the detection of RNA. NASBA is an established diagnostic tool in clinical use, with a theoretically bigger analytical sensitivity than reverse RT-PCR for pathogens detection, but is not progressing toward implementation in food analysis. This is unfortunate, because it has a potential for detection of viable cells through selective amplification of messenger RNA, even in a background of genomic DNA, which PCR does not possess. However, in some instances, an unexpected amplification of genomic DNA was observed using the NASBA technique. The availability of methods for rapid, sensitive, and selective detection of viable microbial pathogens in foods is a goal worth pursuing, and a developmental effort to explore and capitalize on NASBA’s potential in this regard could be worthwhile [45].

## 4. Gene Chip Technology

GC technology was first proposed for biological detection more than 30 years ago (Southern et al. 1989) [46]. This technology uses the principle of nucleic acid hybridization to detect the gene of the sample. A large number of DNA or RNA fragments are hybridized on the surface of the solid carrier in the way of base pairing, and then the hybridization signal and intensity are detected by the scanning system, so as to obtain the gene information of the sample and realize qualitative and quantitative, as shown in Figure 3.

GC fixed the probe molecule on the surface of the chip, and then hybridized with the marked sample molecule. The sample molecular data were obtained by monitoring the strength of the hybridization signal to distinguish the bacterial species. In recent years, the research on the detection of foodborne pathogens by GC gradually increased. GC is the first developed, the earliest in research and development, and the most used technology in biochip technology. In the detection process of foodborne pathogens, this technology can effectively distinguish living bacteria.

EOM et al. (2007) [44] achieved the simultaneous detection of seven strains, such as *Shigella* by 16S rDNA-based gene chip technology, but the disadvantage is that false positive results very easily appear in the process of multiple detections of actual samples. Researchers used gene chip technology to comprehensively compare the whole genome sequences of 11 species of *Campylobacter jejuni*, and obtained specific indicators of the pathogenicity of *Campylobacter jejuni*, which effectively verified the biological characteristics of the strain [45].

GC is a highly specific and sensitive molecular technology that can process samples at a high throughput and detect many pathogens at the same time. The technology is simple to operate and produces accurate data rapidly. However, the equipment is expensive and it requires a high level of skill to operate. GC preparation and the hybridization process to detect relatively few pathogens is time-consuming, costly, and has poor repeatability [46,47]. Therefore, the food testing industry needs to expand the application potential of GC through technological research and development, and better ensure the safety of food production and processing from two aspects: the detection of food raw materials and the detection of pathogenic microorganisms.

## 5. Gene Probe Technology

Gene probe technology was first proposed more than 45 years ago (Southern 1975). This technology depends on being able to make a probe using the DNA sequence of a known gene. When that sequence comes into contact with denatured single-stranded genomic DNA, and if the bases of the two are completely paired, they complement each other to form a double strand, and thereby show that the genomic DNA tested contains a known gene sequence. A complementary DNA single strand can be combined into a double strand under certain conditions. This combination is specific and is carried out according to base complementarity not only between DNA and DNA, but also between DNA and RNA. There are two necessary conditions for gene detection: one is the necessary specific DNA probe; and the second is essential genomic DNA. When both are denatured into a single chain state, molecular hybridization occurs [48].

Gene probe technology uses the characteristics of gene denaturation and repeatability to compare and study the gene sequences in food, so as to ensure the safety of food. In the current gene probe technology, two methods are mainly used: heterogeneous hybridization and in-phase hybridization. These two methods are based on gene probes to detect *Staphylococcus aureus*, *Salmonella,* and other bacteria harmful to human health in food to judge whether the quality of food meets the standards of the relevant departments [49]. 

Gene probe technology was used to rapidly detect a variety of foodborne pathogens for many years [50,51]. This technology, when combined with PCR, can detect many pathogen species simultaneously. However, the non-specific fluorescent probe can interact with complex components present in food, leading to inaccurate results. That limitation can be overcome by selecting biological molecules that have a high degree of specificity.

## 6. Summary

Although the traditional detection method of foodborne pathogens is sensitive enough, it is often too time-consuming for practical use, and it takes a few days to a week to complete. Therefore, a new method to overcome this performance limitation is needed. Recently, several methods for rapid detection of foodborne pathogens were explored and developed. However, most of them still need to improve sensitivity, selectivity, or accuracy to be used for any practical purpose. We summarized several various detection methods, as shown in the Table 3.

Nucleic acid-based methods have high sensitivity and require less time to detect foodborne pathogens and toxins than traditional culture-based techniques, but most of them require trained personnel and expensive instruments, which limits their use in the actual environment. The rapid and accurate detection of foodborne pathogens harmful to humans in the food chain is an important first step toward controlling those pathogens and then eliminating them.

Molecular technology was used for almost 50 years to detect the most common foodborne pathogens. Although it can rapidly detect specific species of pathogens present at low levels in food, the specialized equipment needed to undertake the tests can be expensive, often the gene sequencing of the target pathogens was completed, which limits the use of the technology, and some, such as mPCR and RT-qPCR, require a high level of operational skill. Others, such as the LAMP, are faster than other molecular technologies at detecting and identifying soilborne pathogens accurately and reliably. The accuracy and sensitivity of the technology for rapidly detecting foodborne pathogens will inevitably improve, and the detection cost will reduce to the point where it can be applied routinely as part of a foodborne pathogen surveillance programme. In recent years, biosensors were developed as new diagnostic methods to minimize the limitations of common pathogen detection methods. However, the results of biosensors can be false negatives by using phage or plasmid host ranges that are either too specific or too extensive. Therefore, a reliable, accurate, fast, simple, sensitive, selective, and cost-effective detection method will be ideal. This pathogen detection method will provide great commercial advantages in the food industry and related fields. Moreover, the trend across various methods will produce new devices or methods to enhance the advantages of rapid detection methods.

Our research provides, as a basis for further exploring other foodborne pathogen detection and identification, methods that have the potential to be developed for that purpose. High-throughput, cost-effective, low-skill methods have yet to be developed for many of the common foodborne pathogens. Molecular technologies will need to be continuously improved to reduce the risk of foodborne pathogens affecting human health. In conclusion, there are many promising applications in the field of rapid and automated detection of foodborne pathogens. In view of the wide applicability and great potential of these methods, there is still a great opportunity for further development in the near future.

## Figures and Tables

**Figure 1 molecules-27-08262-f001:**
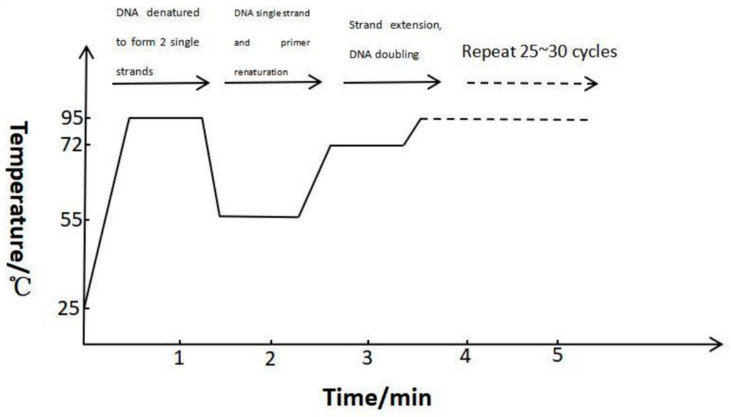
The process diagram of mPCR. Firstly, DNA denatured to form two single strands at 95 °C. Then, DNA forms a single strand and primer renatures at 55 °C. Next, strand extension and DNA doubling at 72 °C. Finally, whole process repeats 25~30 cycles.

**Figure 2 molecules-27-08262-f002:**
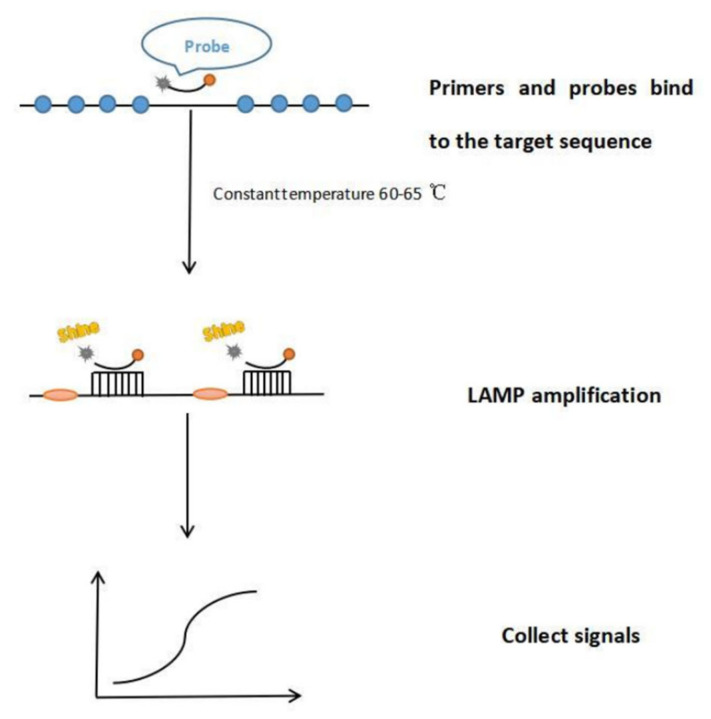
The process diagram of LAMP. Step 1, bind primers and probes to the target sequence. Step 2, amplify LAMP. Step 3, Collect signals.

**Figure 3 molecules-27-08262-f003:**
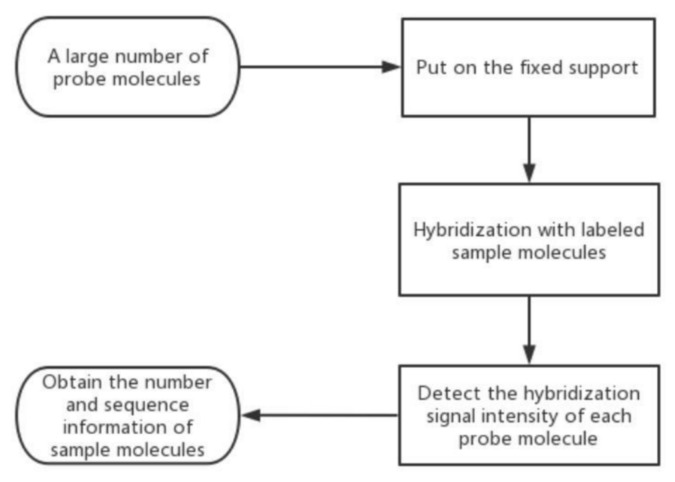
The schematic diagram of GC. At first, we need to collect a large number of probe molecules. Secondly, we put them on the fixed support, and hybridize them with labeled sample molecules. Thirdly, we should detect the hybridization signal intensity of each probe molecule. At last, we can obtain the number and sequence information of sample molecules.

**Table 1 molecules-27-08262-t001:** Application of PCR technology in the detection of foodborne pathogens.

Detection Methods	Detected Pathogens	Detected Food	References
mPCR	*Escherichia coli*	Foodborne pathogenic bacteria	[13]
	*Listeria monocytogenes*, *Listeria ivanovii*, *Listeria innocua*, and atypical *L. innocua*	Green romaine samples	[15]
	*Vibrio parahaemolyticus*	fresh water fish and shellfish	[17]
	*Listeria monocytogenes*	Fruit, vegetables and sprouts retailed in the Czech Republic	[30]
	*Salmonella*	Fresh vegetables in perak, Malaysia	[31]
RT-qPCR	*Shigella* toxin	Beef and sprout enrichment cultures	[22]
	*Salmonella*	Poultry facilities	[23]
	*Listeria monocytogenes*	Meat	[24]
	*Listeria monocytogenes*	60 animal, aquatic and dairy products	[25]
	*Salmonella*	Cooked ham	[26]
	*Staphylococcus aureus*, *Salmonella* and *Shigella*	Fresh pork	[27]
	*E. coli* O157:H7	Luria-Bertani broth	[28]

**Table 2 molecules-27-08262-t002:** Application of loop-mediated isothermal amplification in the detection of foodborne pathogens.

Techniques	Detected Pathogens	Detected Food	References
LAMP	*stxA_2_*	*Escherichia coli* O157:H7 cells	[34]
	*Listeria monocytogenes* et al.	Dairy products	[35]
	*Salmonella* strains	Various food samples	[36]
NASBA	*Chlamydophila pneumoniae*	HEp-2 cells	[43]
	*Aspergillus fumigatus*	*Clavibacter michiganensis* subsp. *sepedonicus* strain	[44]

**Table 3 molecules-27-08262-t003:** Characteristics and shortages of various detection methods.

Types	Characteristics	Shortages
PCR	The process is rapid and sensitive.	PCR methods are not able to differentiate between the live and dead cells. There are chances of generating a false positive signal due to binding to non-specific double-stranded DNA sequences.
mPCR	It has high specificity and sensitivity, and can detect multiple pathogenic bacteria at the same time.	Affected by PCR inhibitors, the amplification efficiency is low, the primer design is difficult, and it is impossible to distinguish dead bacteria from live bacteria.
RT-qPCR	It does not need post-processing of amplification, and real-time monitoring of amplification products is required.	The cost is high, and it will be affected by inhibitors, and requires professional operation.
LAMP	It has easy operation and low cost.	Primer design is difficult, easy to cross contaminate, and high false positives.
NASBA	Rapid response, simple operation and low cost.	Large volume detection cannot be carried out, the sample preparation is complex, and the sample must be a living organism.
GC	It can realize high-throughput and parallel detection of pathogenic bacteria in food, and the operation is simple and rapid.	Chip preparation and hybridization are time-consuming and costly, requiring professional operations.
Gene probe technology	It can detect multiple pathogens at the same time.	The non-specificity of the fluorescent probe may cause distortion of the results.
Whole genome sequencing (WGS)	Establish multiple libraries for representative individuals of different varieties to conduct whole genome de novo assembly sequencing.	Need to consume more computing resources and time.
Multilocus sequence typing (MLST)	Fast, simple, good repeatability, and high discrimination.	Not suitable for organisms with frequent recombination.
Metagenomics	It does not depend on the separation and culture of microorganisms, thus reducing the bottleneck problems caused thereby.	The method of sample extraction needs to be improved and bioinformation analysis depends on the complexity of the sample.
Biosensors	Faster, more cost effective, easy to carry out, and less labor-intensive.	The results of biosensors can be false negatives.

## Data Availability

All data are contained within the manuscript.

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
