# Peer review of "Molecular Methods for Identification and Quantification of Foodborne Pathogens"

_molecules, 2022, doi:10.3390/molecules27238262_

Round 1

Reviewer 1 Report

I have read the review entitled “Molecular methods for identification and quantification of foodborne pathogens” thoroughly.

The review is well organized and concisely written. It helps to compare between the currently used molecular methods for identification and illustrating advantages and disadvantages for all methods.

I have some few issues regarding the mis-writing issues that should be taken in consideration.

For example;

Line 24: please write food outbreaks instead of outbreaks of disease.

Line 31: Please write: According to WHO, foodborne pathogens…..

Line 35: Campylobacter Spp. also should be included in foodborne pathogens.

Line 38: Italy and Sweden are also EU members.

Line 44: please write: are spreading instead of have spread

Line 57: PCR is always specific to a particular DNA sequence not for a whole pathogen.

Table 2: please revise Ref.40.

My main issue with the review is that authors should also included WGS, MLST and metagenomics as up to date molecular techniques for identification of pathogens with discussing advantages of rapidness and accuracy, in addition to disadvantages of high cost mainly.

So, I recommend to discuss the above-mentioned techniques in the review, and resubmit it again.

Sincerely,

Author Response

Point 1: Line 24: please write food outbreaks instead of outbreaks of disease.

Response 1: Thank you for your comments. We have wrote “food outbreaks” in this sentence.

Point 2: Line 31: Please write: According to WHO, foodborne pathogens......

Response 2: Thank you for your comments. We have made change.

Point 3: Line 35: Campylobacter Spp. also should be included in foodborne pathogens.

Response 3: Thank you for your comments. We have added “Campylobacter Spp.” to this sentence.

Point 4: Line 38: Italy and Sweden are also EU members.

Response 4: Thank you for your comments. We have merged them to keep them right.

Point 5: Line 44: please write: are spreading instead of have spread.

Response 5: Thank you for your comments. We have made change.

Point 6: Line 57: PCR is always specific to a particular DNA sequence not for a whole pathogen.

Response 6: Thank you for your comments. We have modified it.

Point 7: Table 2: please revise Ref.40.

Response 7: Thank you for your comments. Ref.40. has been revised.

Point 8: My main issue with the review is that authors should also included WGS, MLST and metagenomics as up to date molecular techniques for identification of pathogens with discussing advantages of rapidness and accuracy, in addition to disadvantages of high cost mainly. So, I recommend to discuss the above-mentioned techniques in the review, and resubmit it again.

Response 8: Thank you for your comments. In “Summary”, we have added these methods to compare their advantages and disadvantages by accessing to information. We think it is more intuitive to list them in the table than to introduce them one by one, so, we hope to get your approval.

Reviewer 2 Report

General Comments

The authors briefly described 7 commonly used molecular methods to detect and quantify important foodborne pathogens. The authors only list who applied which methods to detect what pathogens. There were some discussions regarding the advantages and disadvantages of each technique. However, not much new information was presented in this review article.  

Figures: there were 3 figures in the article, but they were all labeled as Figure 1. In addition, the figures were so small to observe, and the figure legends were too brief.

Tables: Table 2, why [Error! Reference source not found. 40] appeared here?  

Reference: What do the [D] in line 308, [J] in line 313, [M] in line 314 stand for? And many places in the reference list had the same situation. Line 345, and 359, there are no article titles. Line 310 and 354, why were the authors' names capitalized for each letter? The authors must reformat the reference again to fit the journal's style.

Author Response

Point 1: Figures: there were 3 figures in the article, but they were all labeled as Figure 1. In addition, the figures were so small to observe, and the figure legends were too brief.

Response 1: Thank you for your comments. Firstly, we have changed the 3 figures as Figure 1, Figure 2 and Figure 3. Secondly, we have revised the size of figures according to the template. Thirdly, we have added the explanation for every figure.

Point 2: Tables: Table 2, why [Error! Reference source not found. 40] appeared here?

Response 2: Thank you for your comments. We have revised the Ref.40.

Point 3: Reference: What do the [D] in line 308, [J] in line 313, [M] in line 314 stand for? And many places in the reference list had the same situation. Line 345, and 359, there are no article titles. Line 310 and 354, why were the authors' names capitalized for each letter? The authors must reformat the reference again to fit the journal's style.

Response 3: Thank you for your comments. We have reformated the references according to the template.

Reviewer 3 Report

In the review, Authors focus on molecular method applied in foodborne field to detect pathogens. 

The review is quite organized however, sometime, they do not thoroughly analyze the method described.

I suggest to Authors the following revisions before to publication of the manuscript.

Major revision

The bibliography could be implemented with more recent papers in that only 17% of citated publications belong to the last 6 years. 

In “introduction” such as in “summary” the concept of the need to applied molecular method in this field could be highlighted. For example, comparing molecular methods with proposed recently biosensors (DOI: 10.3390/nano10030501; DOI: 10.3390/bios10060058).

The Authors described in their manuscript that: “Although, the traditional detection method of foodborne pathogens is sensitive enough, they are time-consuming”. For this region, I suggest to Authors to indicate for each described molecular method the detection limit of the target pathogens and the time required to have a result.

Minor revision

All the picture could be increased in the resolution such as in the text font size.

Author Response

Point 1: The bibliography could be implemented with more recent papers in that only 17% of citated publications belong to the last 6 years. 

Response 1: Thank you for your comments. We have consulted relevant materials and several references were added or replaced, such as Ref.30, Ref.31 and so on.

Point 2: In “introduction” such as in “summary” the concept of the need to applied molecular method in this field could be highlighted. For example, comparing molecular methods with proposed recently biosensors (DOI: 10.3390/nano10030501; DOI: 10.3390/bios10060058).

Response 2: Thank you for your comments and references provided. We have added the comparison of molecular methods and biosensors, and listed their advantages and disadvantages to highlight the significance of applying molecular methods.

Point 3: The Authors described in their manuscript that: “Although, the traditional detection method of foodborne pathogens is sensitive enough, they are time-consuming”. For this region, I suggest to Authors to indicate for each described molecular method the detection limit of the target pathogens and the time required to have a result.

Response 3: Thank you for your comments. The detection limit of partial pathogens have been listed in the text. Due to different sample sizes, different sample adding times and other uncontrollable factors, there would be no way to give a specific time to have a result. Generally, the reaction time of PCR is about 2-4 hours, the reaction time of “Loop-mediated isothermal amplification” is 30-60 minutes, the reaction time of “Nucleic acid sequence-based amplification” is about 2 hours, the reaction time of “GC” is about half an hour and the reaction time of “Gene probe technology” is about 4-5 hours.

Point 4: Minor revision: All the picture could be increased in the resolution such as in the text font size.

Response 4: Thank you for your comments. We have made changes according to the template.

Round 2

Reviewer 1 Report

Dear all,

hereby I accept the current version of the manuscript to be published as the authors have addressed all mentioned points and have edited the manuscript thoroughly.

Reviewer 2 Report

The authors had revised the manuscript according to the previous suggestions. Still, I list some specific comments as follows:

Line 125-126: RT-qPCR has become one of the most preferred methods…, delete “a”

Line 375-393: I suggest the authors rephrase the figure legend. The passive tense is preferred in scientific writing.

Line 457: , it is rather time-consuming for practical use…